# Effect Analysis of Hydrogen Peroxide Using Hyperspectral Reflectance in Sorghum [*Sorghum bicolor* (L.) Moench] under Drought Stress

**DOI:** 10.3390/plants12162958

**Published:** 2023-08-16

**Authors:** Ki Eun Song, Se Sil Hong, Hye Rin Hwang, Sun Hee Hong, Sang-in Shim

**Affiliations:** 1Department of Plant Life Science, Hankyong National University, Ansung 17579, Republic of Korea; qjrn00@naver.com (K.E.S.); shhong@hknu.ac.kr (S.H.H.); 2Department of Agronomy, Gyeongsang National University, Jinju 52828, Republic of Korea; hss331@hanmail.net (S.S.H.); hyerin999@naver.com (H.R.H.); 3Institute of Life Sciences, Gyeongsang National University, Jinju 52828, Republic of Korea

**Keywords:** climate change, water stress, *Sorghum bicolor*, vegetation index, photosynthesis

## Abstract

Due to global climate change, adverse environments like drought in agricultural production are occurring frequently, increasing the need for research to ensure stable crop production. This study was conducted to determine the effect of artificial hydrogen peroxide treatment on sorghum growth to induce stress resistance in drought conditions. Hyperspectral analysis was performed to rapidly find out the effects of drought and hydrogen peroxide treatment to estimate the physiological parameters of plants related to drought and calculate the vegetation indices through PLS analysis based on hyperspectral data. The partial least squares (PLS) analysis collected chlorophyll fluorescence variables, photosynthetic parameters, leaf water potential, and hyperspectral reflectance during the stem elongation and booting stage. To find out the effect of hydrogen peroxide treatment in sorghum plants grown under 90% and 60% of field capacity in greenhouses, growth and hyperspectral reflectance were measured on the 10th and 20th days after foliar application of H_2_O_2_ at 30 mM from 1st to 5th leaf stage. The PLS analysis shows that the maximum variable fluorescence of the dark-adapted leaves was the most predictable model with R^2^ = 0.76, and the estimation model suitability gradually increased with O (R^2^ = 0.51), J (R^2^ = 0.73), and P (R^2^ = 0.75) among OJIP parameters of chlorophyll fluorescence analysis. However, the estimation suitability of predictions for moisture-related traits, vapor pressure deficit (VPD, R^2^ = 0.18), and leaf water potential (R^2^ = 0.15) using hyperspectral data was low. The hyperspectral reflectance was 10% higher at 20 days after treatment (DAT) and 3% at 20 DAT than the non-treatment in the far red and infra-red light regions under drought conditions. Vogelmann red edge index (VOG REI) 1, chlorophyll index red edge (CIR), and red-edge normalized difference vegetation index (RE-NDVI) efficiently reflected moisture stress among the vegetation indices. Photochemical reflectance index (PRI) can be used as an indicator for early diagnosis of drought stress because hydrogen peroxide treatment showed higher values than untreated in the early stages of drought damage.

## 1. Introduction

Sorghum (*Sorghum bicolor* L. Moench) is a member of Poaceae and is one of the top five food crops along with wheat, rice, corn, and barley [1], and has a high value not only as food but also as feed or sugar crop [2]. In addition, considering that the decline in crop growth caused by increasing drought stress due to climate change threatens food production [3], sorghum, which grows well in arid and semi-arid regions in West Africa, is of great importance in the region threatened by drought. Under the unfavorable agricultural production environment, research for sorghum to ensure a stable crop yield is increasing for food production in the world [4,5].

Image analysis techniques, including remote sensing that is actively introduced into agriculture to recognize crop reactions and damage due to environmental changes rapidly, is known as a tool to estimate various variables reflecting crop growth and physiological characteristics accurately [6,7,8]. In terms of utilization, this technology has attracted significant attention in crop monitoring and predicting crop yields [6], as it can non-destructively and relatively inexpensively characterize crop conditions [9]. Compared to multi-spectral spectrum systems that acquire images in limited several spectral bands in remote sensing and sensing technologies, hyperspectral spectrum sensing technology that acquires images in narrow and continuous spectral bands provides continuous spectra for images [8,10]. Therefore, hyperspectral spectral data are more sensitive to specific crop variables, and reflectance in wide ranges has the advantage of providing various information such as nitrogen and chlorophyll content in leaves [11], LAI [12], and crop yield prediction [7,13] using normalized difference vegetation index between bands. However, unlike the use of average spectral reflectance in a wide range band, there is also a disadvantage that using a specific narrow band may lead to the loss of particular information [8,14,15]. Partial least square (PLS) analysis, used to analyze hyperspectral data, is a method of easily processing a data matrix in which many variables, such as hyperspectral reflectance data, describe each object. This technology can extract relevant parts of information about huge data matrices and generate the most reliable models for other calibration methods [16,17]. The PLS method has been used in plant science to predict plant biomass [8], LAI, nitrogen [10], and chlorophyll concentrations [18]. The relationship between spectral properties and reflectance to the functional characteristics of plants is not yet clear or known, and the spectral reflectance of plants related to plant water stress or physiological changes is very complicated. However, spectral vegetation indices are analytical tools that can evaluate the parameters related to the physiology, morphology, and biochemical characteristics of plants using several wavelengths to understand vegetation characteristics. Therefore, it is very useful as a tool for evaluating plant growth under various environmental conditions [19].

The stresses that inhibit normal growth induce the generation of reactive oxygen species (ROS) in plants, causing damage, and plants become resistant to stress through antioxidant responses to ROS. In this process, ROS acts as a signaling substance for inducing stress resistance, and the interest in the practical application of artificial induction of ROS for coping with stress is also growing. Hydrogen peroxide (H_2_O_2_) is a representative ROS substance that improves resistance to various abiotic stresses through signal transduction in plants. It has been proven to have its beneficial effects partially in some physiological processes like seed germination [20,21,22,23]. If the external treatment of hydrogen peroxide is effective in ameliorating abiotic stress, hydrogen peroxide treatment is of great practical value in adverse environmental conditions that become more serious to plants by increasing the risk of stress like drought due to climate change because H_2_O_2_ is easy to apply to plant leaves by spraying [24,25]. On the other hand, empirical studies on the effect of artificially applied H_2_O_2_ on the growth of crops during the entire growth period are still insufficient.

This study was conducted to know the validity of the non-destructive measurement method for determining the effect of hydrogen peroxide treatment on sorghum in drought conditions. The response of plants was confirmed through vegetation indices based on hyperspectral reflectance data and physiological variable estimation through the PLS model.

## 2. Results

### 2.1. PLS Analysis Using Physiological Traits and Hyperspectral Data of Sorghum

Partial least squares (PLS) regression analysis was conducted to increase the utilization of hyperspectral analysis in field conditions by analyzing the relationships between hyperspectral reflectance data and leaf chlorophyll fluorescence parameters, photosynthetic variables, and leaf water potential (Figure 1, Figure 2 and Figure 3). PLS analysis based on the physiological parameters measured during vegetative growth and hyperspectral data showed different accuracy depending on parameters.

The PLS models of chlorophyll fluorescence variables were divided into predictive models with a high coefficient of determination (R^2^ > 0.5) and models with a low R^2^ (<0.5). Prediction models that showed a relatively high R^2^ (>0.5) were found in models for the parameters of O, T100, K, J, I, P (Fm), S, M, Fo, Fm, Fv, Fv/Fo, ETo/CS, and area in the chlorophyll fluorescence responses. The chlorophyll fluorescence parameters represented by the prediction model with low R^2^ (<0.5) were tFm, Vj, PI, Mo, DIo/CS, DIo/RC, RC/CSo, RC/CSm, and ETo/ABS (Figure 1). In each prediction model, the PLS models for maximum fluorescence intensity (Fm) and the maximum variable fluorescence (Fv = Fm − Fo) of dark-adapted leaves showed the highest R^2^ with 0.75 and 0.76, respectively, and the coefficient of determination of tFm was 0.09, indicating that the estimated suitability was low. Among the parameters from OJIP analysis of chlorophyll fluorescence, the coefficients of determination of variables related to energy flow (ETo/ABS, DIo/RC) at the PSII reaction center showed low R^2^ of 0.21 and 0.12, respectively (Figure 1).

In the PLS model for photosynthesis-related variables and leaf water potential, photosynthetic rate (Pn), transpiration rate (E), and stomatal conductance (C) also showed relatively higher R^2^ (Figure 2), but vapor pressure deficit (VPD) and leaf water potential showed low coefficients of determination, 0.18 and 0.15, respectively (Figure 2 and Figure 3).

### 2.2. Changes in Hyperspectral Reflectance and Vegetation Indices by Hydrogen Peroxide Treatment

This research investigated the effect of hydrogen peroxide treatment on sorghum seedlings under drought conditions using 35 vegetation indices (see Table 1) based on hyperspectral analysis. The hyperspectral reflectance of sorghum leaves in the visible light range (400–700 nm) by hydrogen peroxide treatment differed between well-watered and drought conditions (Figure 4). The overall hyperspectral reflectance on leaves was increased by hydrogen peroxide 10 DAT but did not show a great difference. Even 20 DAT, there were no significant differences in the visible light range between the H_2_O_2_ treatment and control under well-watered conditions. However, in the drought condition, the reflectance of the blue light (460–500 nm) and red light (620–680 nm) band in the visible light range was higher in hydrogen peroxide treatment than that of the control plants 10 DAT, but lower in green light (500–570 nm). The reflectance in the visible range of plants treated with hydrogen peroxide was higher than that of untreated plants at 20 DAT (Figure 4).

The reflectance in the far-red region of 700–800 nm was similar between hydrogen peroxide treatment and control under well-watered conditions at 10 and 20 DAT. The reflectance in the far-red region was higher by 10% at 10 DAT in hydrogen peroxide treatment than that of the control. The difference was reduced to 3% at 20 DAT (Figure 4). Similar results were observed in the reflectance in the near-infrared region (>800 nm) over time under drought conditions (Figure 4).

As a result of calculating the vegetation indices of sorghum for each treatment based on reflectivity (Table 1) it was confirmed that the vegetation indices were higher in hydrogen peroxide treatment than in non-treatment 10 DAT (Table 1). The effect of hydrogen peroxide treatment showed significant differences in the vegetation index GNDVI, VOG REI 1, PSSR b, PSSR c, GCI, PRI*CI, and ARI 2 under the well-watered conditions on the 10th day of treatment. However, it was confirmed that the effect of hydrogen peroxide treatment was significantly different only in PRIM3, HI-2013, and ARI 1 under the well-watered and drought conditions at 20 DAT. In particular, the three PRIs, PRIM3 using the reflectance at 670 nm reflect well the influences of soil moisture conditions and hydrogen peroxide treatment, and among the vegetation indices related to carotenoids, PSSRc and PRI*CI were good indices to know the effects of soil moisture conditions and hydrogen peroxide treatment. Among the vegetation indices related to photosynthetic pigments, PSSRb adequately reflected the effect of hydrogen peroxide under different soil moisture conditions. The response of the vegetation indices, in general, was better in drought conditions than in well-watered conditions for the effects of hydrogen peroxide. In addition, among the 35 vegetation indices, SIPI, SRPI, NPQI, and SR 400/900, which use reflectance in the 400–450 nm, did not show any effect by treatment timing, soil water conditions, and hydrogen peroxide treatment.

## 3. Discussion

### 3.1. PLS Analysis Using Physiological Traits and Hyperspectral Data of Sorghum

The maximum fluorescence intensity (Fm) was highly suitable for using hyperspectral data. Still, the t Fm, which represents the time to reach the maximum fluorescence intensity, was less suitable for the predictive model. According to Strasser et al. [26], chlorophyll fluorescence kinetics can be divided into O-J, J-I, and I-P stages. J-I is related to the donor showing the water-splitting activity of photosystem II. This stage reflects the temperature effect on fluorescence induction, and the water-splitting activity is sensitive to temperatures between 35–40 °C [27,28]. Because it is not an estimation model that knows the effect of temperature in this research, the PLS model for the J-I stage was not highly suitable. In a study by Sobejano-Paz et al. [29], PLS analysis using the hyperspectral reflectance measured in drought-treated soybeans and maize showed high coefficients of 0.92, 0.87, and 0.82, respectively, as shown in the results of this study. On the other hand, since we used the reflectance of the range 400 to 900 nm in this research, the suitability of the predictive model for water-related traits (VPD, leaf water potential) was to be low due to a lack of reflectance data in the infrared range reflecting water status.

As previous research shows, hyperspectral analysis has a highly accurate estimation for chlorophyll fluorescence [30]. Our results also showed high estimation in OJIP parameters related to maximum fluorescence intensity and maximum variable fluorescence. Hyperspectral data are easily obtained from large areas with nondestructive methods for plants growing in the field. Therefore, it can be used as an efficient evaluating tool for crop management if used for variables with high coefficients of determination for photosynthesis-related traits.

### 3.2. Changes in Hyperspectral Reflectance and Vegetation Indices by Hydrogen Peroxide Treatment

The vulnerability of crops to water stress resulted in an alteration of crop growth and changes in the leaf structure or water status of the crop [31,32]. In addition, visible symptoms under drought conditions generally include leaf rolling or folding, leaf withering, leaf discoloration, and canopy opening [31]. In temporary drought, crops act a photoprotection through heat and fluorescence emission to protect plant tissues, and in continuous drought stress, changes in the sunlight reflectance on the leaf surface, light absorption rate, and chloroplast degradation in leaves [33,34]. The CIR of plants treated with hydrogen peroxide was higher than that of the control. Therefore, this index is considered a helpful vegetation index reflecting the effect of hydrogen peroxide treatment according to water deficit conditions. In addition, the vegetation index using the same wavelengths (710 and 750 nm) used for CIR calculation is RE-NDVI, and it was higher in the hydrogen peroxide treatment than the control during the initial period of drought conditions. The results of Zhang and Zhou [35] also showed that RE-NDVI is an index sensitive to changes in water status and drought stress among the vegetation indices in maize grown under different soil moisture conditions. Normalized difference vegetation index (NDVI), the most commonly used vegetation index among spectral vegetation indications, reflects sensitively to vegetation biomass but is relatively less sensitive for physiological conditions [36]. However, in this study, NDVI was a helpful index as it was higher in the hydrogen peroxide treatment than the control during the initial stage of drought stress.

The photochemical reflection index (PRI), which is used to determine photosynthesis efficiency in connection with the xanthophyll cycle, is a valuable indicator of the physiological condition and stress level of crops [36,37]. Thénot et al. [38] reported that the PRI of leaves with low leaf water potential (−0.48 MPa) was higher than that of leaves with high leaf water potential (−0.38 MPa) in quinoa and Alordzinu et al. [19] also reported that the PRI index of tomatoes grown under different soil moisture conditions tends to decrease when stressed.

Remote sensing techniques are widely used in vegetation management as a non-destructive method to estimate the biochemical and morphological properties based on energy absorption and scattering at different spectral ranges [39,40]. Photosynthesis and moisture movement characteristics are obtained through optical or thermography analysis. However, the use of remote sensing techniques to measure plant function and physiological features is complicated because the mechanism of the reflectance and absorption of light related to functional changes in plants is not known accurately [41,42]. Our results also show functional properties of hyperspectral properties for estimating the physiological status of plants under different growth conditions.

The most frequently applied remote sensing technique for monitoring plant responses to moisture stress is the use of vegetation indices [43,44]. In particular, the reflectance analysis of spectral parameters on the surface of a plant canopy is widely used in remote sensing techniques. Vegetation indices such as the NDVI and PSSR are associated with plant structural characteristics such as LAI and pigment conditions such as chlorophyll content [45,46]. However, in the development of plant remote sensing technology, the finding for efficient reflectance is essential because specific reflectance analysis in certain bands that sensitively and efficiently reflect the responses to stress is critical [47].

In our study, PRI was lower in drought conditions than in well-watered conditions, but the hydrogen peroxide treatment did not show a significant difference. However, PRIm3 and PRI*CI indices showed substantial differences in the effects of drought and hydrogen peroxide treatment in our results, so PRIm3 and PRI*CI are more efficient indices than PRI. Hernández-Clemente et al. [48] examined the vegetation indices using reflectances at 512, 570, 600, and 670 nm in addition to reflectance at 531 nm. They found that PRI512 was sensitive to stomatal conductance and leaf water potential, while PRI570 was sensitive to structural features related to the canopy. The results imply that the effective wavelength differs according to each trait. This result shows that the efficiency of the spectral bands used for determining the condition of the plant is different depending on the target trait. Vegetation indices that showed the effect of hydrogen peroxide on initial drought stress in sorghum can be used to develop reasonable measures through early detection of water stress in other crops.

The role of H_2_O_2_ in reducing water stress is known to be due to increased plant resistance through signal transduction [49]. However, various results have been reported for the proper concentration of hydrogen peroxide that induces signal transduction for increasing resistance. It depends on the plant species and the timing of treatment, and it is considered that there is a difference in appropriate concentration depending on the treated amount. In this study, plants were treated to the extent that the leaf surface was sufficient with a 30 mM H_2_O_2_. In plants, low concentrations of hydrogen peroxide are present in cellular organs such as chloroplasts, mitochondria, and peroxisomes [49], and about 1-5% of mitochondrial O_2_ consumption leads to H_2_O_2_ production [50]. Therefore, it can be possible that if the concentration increases beyond the common level of hydrogen peroxide in the plant, resistance induction by hydrogen peroxide may occur. Considering the result that the concentration increase in hydrogen peroxide by drought is higher than the well-watered condition [51], the increase in drought resistance by hydrogen peroxide may result from reducing the damage to photosynthesis. Hydrogen peroxide is a substance that exists in plants even in natural conditions, so using low concentrations can be a useful practice to improve plant resistance under drought conditions.

## 4. Materials and Methods

### 4.1. Sorghum Growth and Hydrogen Peroxide Treatment

This experiment was conducted in the greenhouse of Gyeongsang National University from 20 March to 10 September in 2022. The temperature of the greenhouse was controlled to 30 ± 5 (day)/20 ± 2 (night) during the experiment. The sorghum variety used in the experiment was cv. Nampungchal, which is short in plant height with middle–late maturity. Plants were sown in ports (14 cm H × 14 cm D) filled with horticultural nursery soil and loam at a 3:1 ratio. The moisture conditions in the pot were set to 80% and 60% of field capacity for well-watered and drought conditions, respectively, based on our preliminary experiment to know the threshold of water stress. The soil water potential was measured daily using a dewpoint potentiometer (WP4-C, Decagon, Pullman, WA, USA), and an appropriate amount of water was irrigated based on this measured soil water potential. To find out the effect of hydrogen peroxide during the early growth stage of sorghum, plants from the 1- to 5-leaf stage were used for the experiment, and 30 mM H_2_O_2_ was sprayed on the same day regardless of the leaf stage. Growth and physiological measurement were conducted on the 10th and 20th days after treatment.

### 4.2. Analysis of Hyperspectral Reflectance and Calculation of Vegetation Indices

Hyperspectral images were collected using a portable hyperspectral camera (Specim IQ, Specim Co., Oulu, Finland) to analyze the hyperspectral reflectance on leaves in sorghum plants. For optimizing images, halogen light (500w × 2) was used as an artificial light source, the distance between the hyperspectral camera and the surface of the plant canopy was 1 m, and the raw hyperspectral image was corrected using a barium sulfate (BaSO_4_) reference plate that has 100% reflectance in the hyperspectral band. Image processing and analysis were performed using the ENVI 5.1 (Exelis Visual Information Solution, Inc. Pearl East Circle Boulder, CO, USA) program. Vegetation indices were calculated using the reflectance at each band from hyperspectral data (Table 2).

### 4.3. Measurement of Chlorophyll Fluorescence, Photosynthesis, and Water Potential

Chlorophyll fluorescence was measured with fully expanded leaves dark-adapted for 30 min using a chlorophyll fluorometer (OS-30p, Opti-sciences, Hudson, NH, USA), chlorophyll fluorescence-related parameters used in this research were described in Appendix A. Photosynthesis-related parameters were measured in the second uppermost leaf from 12:00 to 14:00 using Handheld Photosynthesis System (CID-340, Bio-science, Camas, WA, USA). Leaf water potential was measured using a leaf pressure vessel (Plant water status solutions (3000 Model, Soilmoisture Equipment Crop., Santa Barbara, CA, USA) with a grass sample holder. All measurements of physiological characteristics were conducted for ten plants per replication with four replications.

### 4.4. Statistical Analysis

The average value was compared at the *p* < 0.05 level with Tukey’s test after one-way analysis using the SAS program (ver. 9.4 SAS Inst., Cary, NC, USA). The PLS analysis was conducted by establishing a model using Proc PLS (Algorithm-NIPALS) of the SAS program and selecting the optimal PLS factor that does not differ significantly from the condition with the lowest root mean PRESS (predicted residual error sum of squares) at the *p* < 0.1 level. Spectral reflectance and physiological characteristics were set as independent and dependent variables. The variable importance in projection (VIP) values was calculated for predictors and used as criteria for selecting appropriate spectral bands.

## 5. Conclusions

The maximum variable fluorescence was identified as the most predictable with R^2^ of 0.76, and OJIP parameters were also highly predictable as a result of analyzing PLS models by measuring chlorophyll fluorescence variables, photosynthetic traits, leaf water potential, and hyperspectral reflectance of leaves during culm elongation and booting stage in sorghum. However, the efficiency of prediction for water-related traits like VPD and leaf water potential using hyperspectral data was low. The hyperspectral reflectance was 10% higher in the plants treated with hydrogen peroxide than the control under drought conditions at ten days after treatment and 3% higher at twenty days after treatment. In particular, there was less difference in the far-red and near-infrared regions. The 35 vegetation indices examined in this research showed significant results in water stress. Among the vegetation indices that indicated the ameliorating effects of hydrogen peroxide under the drought conditions, GNDVI, VOG REI 1, PSSRb, PSSRc, GCI, PRI*CI, and ART2 can be used as early indicators for drought stress and recovery. The PRIM3 and PRI*CI, mainly, were sensitive and efficient indices for evaluating water stress.

## Figures and Tables

**Figure 1 plants-12-02958-f001:**
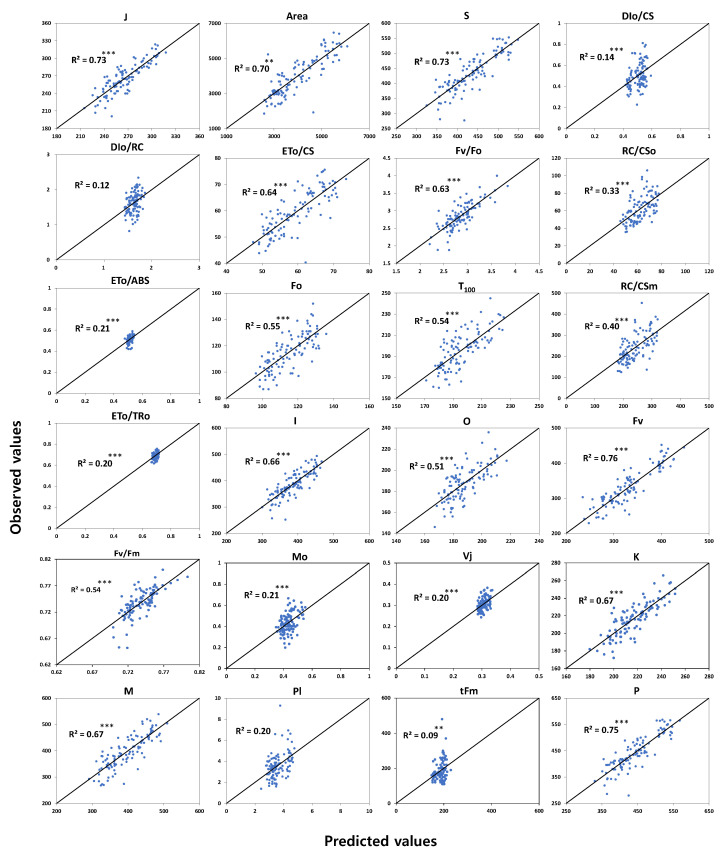
Relationship between observed values and predicted values based on the PLS model using hyperspectral reflectance and chlorophyll fluorescence induction (OJIP test) parameters. ** and *** represent significance level at *p* < 0.01 and *p* < 0.001, respectively.

**Figure 2 plants-12-02958-f002:**
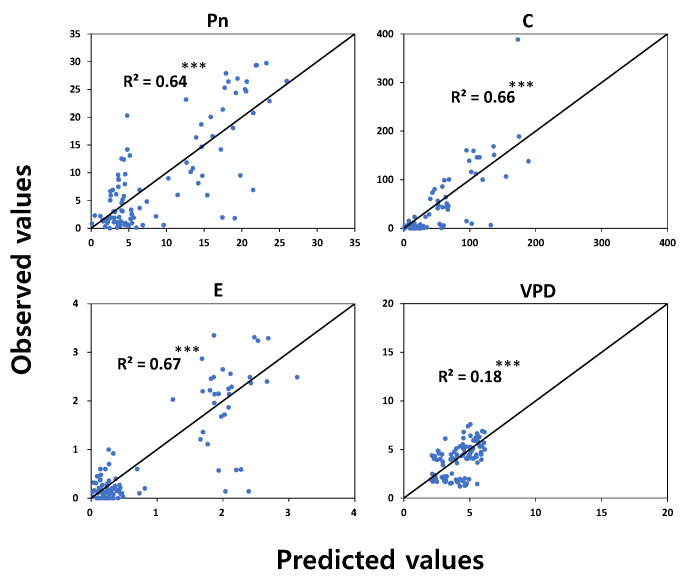
Relationship between observed values and predicted values based on the PLS model using hyperspectral reflectance and photosynthesis parameters. *** represents significance level at *p* < 0.001.

**Figure 3 plants-12-02958-f003:**
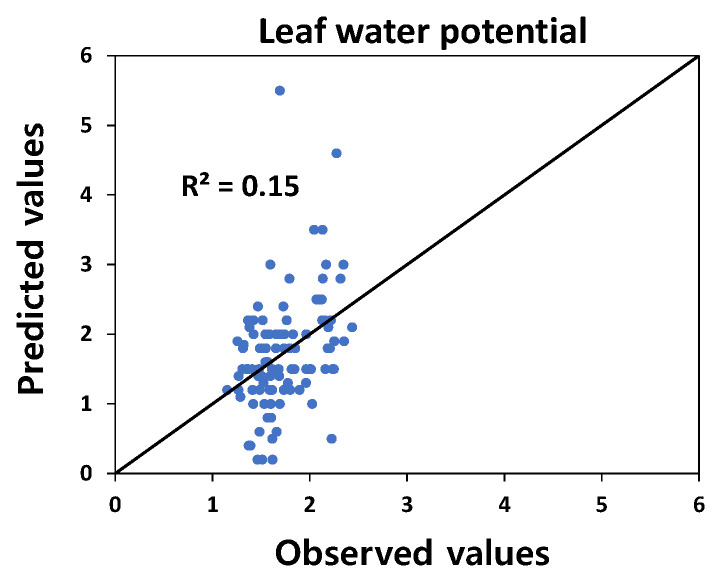
Relationship between observed values and predicted values based on the PLS model using hyperspectral reflectance and leaf water potential.

**Figure 4 plants-12-02958-f004:**
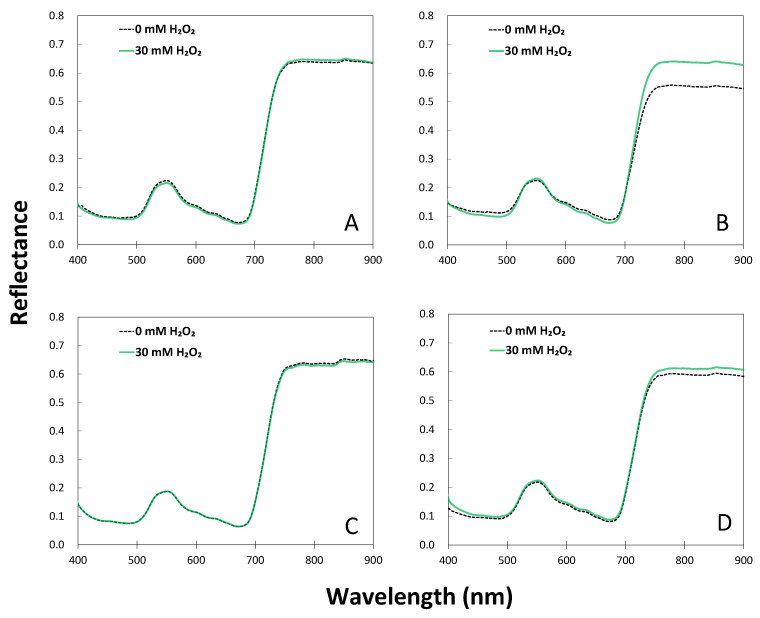
Changes in hyperspectral reflectance by drought and hydrogen peroxide treatments in sorghum seedlings. (**A**,**B**) Indicate the reflectance at 10 DAT, and (**C**,**D**) at 20 DAT.

**Table 1 plants-12-02958-t001:** Changes in vegetation indices by hydrogen peroxide treatments in sorghum seedlings.

Index	10 DAT				20 DAT			
	Well-Watered		Drought		Well-Watered		Drought	
	0 mM H_2_O_2_	30 mM H_2_O_2_	0 mM H_2_O_2_	30 mM H_2_O_2_	0 mM H_2_O_2_	30 mM H_2_O_2_	0 mM H_2_O_2_	30 mM H_2_O_2_
NDVI ^1^	0.780 ^a^*	0.793 ^a^	0.725 ^b^	0.781 ^a^	0.812 ^a^	0.816 ^a^	0.753 ^b^	0.747 ^b^
RE-NDVI ^2^	0.446 ^a^	0.463 ^a^	0.407 ^b^	0.448 ^a^	0.502 ^a^	0.502 ^a^	0.427 ^b^	0.430 ^b^
MRE NDVI ^3^	0.579 ^a^	0.596 ^a^	0.584 ^a^	0.596 ^a^	0.634 ^a^	0.624 ^a^	0.561 ^b^	0.573 ^b^
NRI ^4^	0.416 ^a^	0.420 ^a^	0.373 ^b^	0.430 ^a^	0.413 ^a^	0.430 ^a^	0.393 ^a^	0.377 ^a^
GI ^5^	2.434 ^a^	2.453 ^a^	2.196 ^b^	2.515 ^a^	2.430 ^a^	2.517 ^a^	2.312 ^a^	2.219 ^a^
SRI ^6^	8.132 ^a^	8.540 ^a^	6.175 ^b^	8.030 ^a^	9.912 ^a^	10.123 ^a^	7.185 ^b^	6.924 ^b^
GNDVI ^7^	0.471 ^b^	0.489 ^a^	0.416 ^c^	0.460 ^b^	0.537 ^a^	0.532 ^a^	0.453 ^b^	0.455 ^b^
VOG REI 1 ^8^	1.387 ^b^	1.412 ^a^	1.348 ^c^	1.383 ^b^	1.485 ^a^	1.486 ^a^	1.377 ^b^	1.390 ^b^
PRI ^9^	0.042 ^a^	0.043 ^a^	0.035 ^b^	0.036 ^b^	0.044 ^a^	0.041 ^a^	0.029 ^b^	0.025 ^b^
PSRI ^10^	−0.033 ^a^	−0.030 ^a^	−0.049 ^c^	−0.03 ^b^	−0.022 ^a^	−0.022 ^a^	−0.027 ^a^	−0.027 ^a^
SIPI ^11^	0.966 ^a^	0.964 ^a^	0.940 ^a^	0.949 ^a^	0.968 ^a^	0.972 ^a^	0.974 ^a^	0.972 ^a^
BI ^12^	1.021 ^a^	1.060 ^a^	1.041 ^a^	1.066 ^a^	1.093 ^a^	1.072 ^a^	1.046 ^b^	1.021 ^b^
PSSRa ^13^	8.190 ^a^	8.676 ^a^	6.287 ^b^	8.167 ^a^	9.748 ^a^	9.974 ^a^	7.271 ^a^	7.008 ^a^
PSSRb ^14^	5.945 ^ab^	6.300 ^a^	4.631 ^c^	5.790 ^b^	7.037 ^a^	7.018 ^a^	5.241 ^a^	5.153 ^a^
PSSRc ^15^	6.487 ^ab^	6.928 ^a^	4.840 ^c^	6.233 ^b^	8.054 ^a^	8.190 ^a^	6.022 ^b^	5.969 ^b^
SR 515/570 ^16^	0.758 ^b^	0.754 ^b^	0.790 ^a^	0.744 ^b^	0.753 ^a^	0.738 ^a^	0.749 ^a^	0.750 ^a^
RI ^17^	0.277 ^b^	0.264 ^b^	0.318 ^a^	0.273 ^b^	0.239 ^b^	0.238 ^b^	0.300 ^a^	0.300 ^a^
PhRi ^18^	0.048 ^a^	0.048 ^a^	0.041 ^b^	0.050 ^a^	0.049 ^a^	0.051 ^a^	0.049 ^a^	0.049 ^a^
GCI ^19^	1.860 ^b^	2.008 ^a^	1.475 ^c^	1.766 ^b^	2.423 ^a^	2.367 ^a^	1.716 ^b^	1.756 ^b^
mSR 705 ^20^	3.773 ^a^	3.958 ^a^	3.835 ^a^	3.968 ^a^	4.497 ^a^	4.337 ^a^	3.577 ^b^	3.712 ^b^
CIR ^21^	2.278 ^a^	2.367 ^a^	2.105 ^b^	2.281 ^a^	2.623 ^a^	2.624 ^a^	2.204 ^b^	2.222 ^b^
SRPI ^22^	0.755 ^a^	0.736 ^a^	0.726 ^a^	0.690 ^a^	0.752 ^a^	0.776 ^a^	0.817 ^a^	0.826 ^a^
CCI ^23^	0.364 ^a^	0.369 ^a^	0.319 ^b^	0.367 ^a^	0.352 ^a^	0.358 ^a^	0.329 ^a^	0.315 ^a^
CRI 550 ^24^	3.883 ^a^	4.093 ^a^	3.046 ^b^	3.791 ^a^	4.786 ^a^	5.011 ^a^	3.789 ^b^	3.598 ^b^
PRIm3 ^25^	−0.450 ^b^	−0.455 ^b^	−0.403 ^a^	−0.459 ^b^	−0.449 ^ab^	−0.462 ^b^	−0.417 ^ab^	−0.398 ^a^
PRIm4 ^26^	−0.200 ^a^	−0.200 ^a^	−0.209 ^a^	−0.192 ^a^	−0.202 ^a^	−0.195 ^a^	−0.199 ^a^	−0.201 ^a^
PRI*CI ^27^	−0.114 ^b c^	−0.125 ^c^	−0.076 ^a^	−0.099 ^b^	−0.142 ^b^	−0.134 ^b^	−0.072 ^a^	−0.060 ^a^
RGI ^28^	0.460 ^b^	0.455 ^b^	0.496 ^a^	0.438 ^b^	0.476 ^a^	0.458 ^a^	0.493 ^a^	0.508 ^a^
RARS ^29^	4.682 ^a^	4.932 ^a^	3.747 ^b^	4.580 ^a^	5.577 ^a^	5.632 ^a^	4.384 ^b^	4.368 ^b^
HI_2013 ^30^	0.086 ^ab^	0.092 ^ab^	0.079 ^b^	0.106 ^a^	0.084 ^ab^	0.096 ^a^	0.064 ^ab^	0.054 ^b^
NPQI ^31^	0.083 ^a^	0.074 ^a^	0.047 ^a^	0.066 ^a^	0.101 ^a^	0.088 ^a^	0.058 ^a^	0.063 ^a^
ARI 1 ^32^	−1.349 ^a^	−1.410 ^a^	−1.330 ^a^	−1.536 ^a^	−1.390 ^ab^	−1.541 ^b^	−1.224 ^ab^	−1.137 ^a^
ARI 2 ^33^	2.767 ^ab^	3.007 ^a^	1.627 ^c^	2.487 ^b^	3.745 ^a^	3.855 ^a^	2.584 ^b^	2.520 ^b^
SR 400/690 ^34^	1.320 ^a^	1.440 ^a^	1.317 ^a^	1.474 ^a^	1.613 ^a^	1.425 ^a^	1.232 ^a^	1.314 ^a^
SR 750/700 ^35^	3.620 ^a^	3.797 ^a^	3.149 ^b^	3.661 ^a^	4.186 ^a^	4.224 ^a^	3.368 ^b^	3.348 ^b^

* Means with the same letter within rows are not significantly different (Tukey’s test, *p* < 0.01). ^1^ Normalized difference vegetation index, ^2^ Normalized difference 750/710 red edge NDVI, ^3^ modified normalized difference 705, ^4^ nitrogen reflectance index, ^5^ greenness index, ^6^ simple ratio, ^7^ green normalized difference vegetation index, ^8^ Vogelmann index, ^9^ photochemical reflectance index, ^10^ plant senescence reflectance index, ^11^ structural independent pigment index, ^12^ blue index, ^13^ pigment specific simple ratio (chlorophyll a), ^14^ pigment specific simple ratio (chlorophyll b), ^15^ pigment specific simple ratio (carotenoids), ^16^ simple ratio 515/570, ^17^ redness index, ^18^ physiological reflectance index, ^19^ green chlorophyll index, ^20^ modified red-edge ratio, ^21^ chlorophyll index red edge, ^22^ simple ratio pigment index, ^23^ chlorophyll/carotenoid index, ^24^ carotenoids reflectance index, ^25^ photochemical reflectance index 670, ^26^ photochemical reflectance index 670 and 570, ^27^ carotenoid/chlorophyll ratio index, ^28^ red/green index, ^29^ ratio analysis of reflectance spectra, ^30^ health index, ^31^ normalized phaeophytinization index, ^32^ anthocyanin reflectance, ^33^ anthocyanin reflectance, ^34^ simple ration index, ^35^ simple ration index.

**Table 2 plants-12-02958-t002:** Vegetation indices calculated from hyperspectral reflectance in the experiment.

Index	Formula ^1^	Reference
Normalized difference vegetation index (NDVI)	(R800 − R680)/(R800 + R680)	[52]
Normalized difference 750/710 red edge NDVI (RE-NDVI)	(R750 − R710)/(R750 + R710)	[35]
Modified normalized difference 705 (MRE NDVI)	(R750 − R705)/(R750 + R705 – 2 × R445)	[53]
Nitrogen reflectance index (NRI)	(R570 − R670)/(R570 + R670)	[54]
Greenness index (GI)	R570/R670	[55]
Simple ratio (SRI)	R900/R680	[56]
Green normalized difference vegetation index (GNDVI)	(R750 − R550)/(R750 + R550)	[57]
Vogelmann index (VOG REI 1)	R740/R720	[58]
Photochemical reflectance index (PRI)	(R531 − R570)/(R531 + R570)	[59]
Plant senescence reflectance index (PSRI)	(R680 − R500)/R750	[60]
Structural independent pigment index (SIPI)	(R800 − R445)/(R800 + R680)	[61]
Blue index (BI)	R450/R490	[55]
Pigment specific simple ratio (chlorophyll a) (PSSRa)	R800/R680	[45]
Pigment specific simple ratio (chlorophyll b) (PSSRb)	R800/R635	[45]
Pigment specific simple ratio (carotenoids) (PSSRc)	R800/R500	[45]
Simple ratio 515/570 (SR 515/570)	R515/R570	[62]
Redness index (RI)	R700/R670	[63]
Physiological reflectance index (PhRi)	(R550 − R531)/(R550 + R531)	[64]
Green chlorophyll index (GCI)	(R780/R550) − 1	[65]
Modified red-edge ratio (mSR 705)	(R750 − R445)/(R705 − R445)	[66]
Chlorophyll index red edge (CIR)	(R750/R710)	[67]
Simple ratio pigment index (SRPI)	(R430/R680)	[56]
Chlorophyll/carotenoid index (CCI)	(R531 − R645)/(R531 − R645)	[68]
Carotenoids reflectance index (CRI 550)	(1/R510) − (1/R550)	[69]
Photochemical reflectance index 670 (PRIm3)	(R670 − R531)/(R670 + R531)	[70]
Photochemical reflectance index 670 and 570 (PRIm4)	(R570 − R531 − R670)/(R570 + R531 + R670)	[51]
Carotenoid/chlorophyll ratio index (PRI*CI)	(R531 − R570)/(R531 + R570) × ((R760/R700) − 1)	[59]
Red/green index (RGI)	(R690/R550)	[71]
Ratio analysis of reflectance spectra (RARS)	(R746/R513)	[72]
Health index (HI_2013)	(R534 − R698)/(R534 + R698) − 0.5 × R704	[73]
Normalized phaeophytinization index (NPQI)	(R415 − R435)/(R415 + R435)	[74]
Anthocyanin reflectance (ARI1)	(1/R550 − 1/R700)	[75]
Anthocyanin reflectance (ARI2)	R800(1/R500 − 1/R700)	[69]
Simple ration index (SR400/690)	(R400/R690)	[76]
Simple ration index (SR750/700)	(R750/R700)	[52]

^1^ RXXX means the reflectance value at a certain wavelength (XXX nm).

## Data Availability

The data presented in this study are available on request from the corresponding author.

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
