# Peer review of "Effect Analysis of Hydrogen Peroxide Using Hyperspectral Reflectance in Sorghum [Sorghum bicolor (L.) Moench] under Drought Stress"

_plants, 2023, doi:10.3390/plants12162958_

Round 1

Reviewer 1 Report

The manuscript "Analysis of drought stress and effect of ..." by Song et al. is a methodologically interesting paper on the effect of short drought periods on photosynthesis in Sorghum. The paper reads well,

A few minor questions:

- what do the authors consider a great difference in hyperspectral reflectance? How do they make a difference and where do they draw the line? is there any measure of variation in the measurements ?

- line 227 typo: is using of.... ??

- Material and methods line 258: the authors mention a difference in growth stage; did this come into play as a useful trait in the analysis ? Is there any difference between these stages ?

- maybe add a small paragraph in the M&M about PLS analysis for the readers who are not up to speed

Author Response

Dear Reviewer

Thanks for the valuable comments.

We identified noticeable differences by band from the hyperspectral graph and analyzed them linked to the vegetation index. Statistical difference verification was not performed for the difference in the hyperspectral graph. In addition, we have added the characteristics of hyperspectral analysis to the text.

We have revised the Materials and Methods section according to your comments.

Thank you very much for your good point

Reviewer 2 Report

The part material and methods must be rewritten fro the beginning with some details. The number of plants studied the number of replications are not mentioned in the paper. These informations must be included

Minor editing of English language required

Author Response

Dear reviewer

Thanks for the valuable comments.

We have added the characteristics of hyperspectral analysis to the text. As for the growth stage, plants were at different leaf stages, so plants at 1-5 leaf stages were treated simultaneously regardless of leaf age, but the analysis was performed on plants with the same leaf stage.

Thank you very much for your good point

Reviewer 3 Report

This research is an interesting topic regarding abiotic stress measurement with an easier way to detect as well as conducted how H2O2 can affect drought responses in Sorghum. However, some parts of missing and flaws in the manuscript and needed to be improved. For instance, it was not discussed how hydrogen peroxide treatment can mitigate drought stress in plant physiology and/or biochemistry context with enough references, and also use of abbreviation was not organised well. There are some comments as follows:

L2 I want to suggest some title: Effect of Hydrogen Peroxide Using Hyperspectral Reflectance in Sorghum under drought stress

L3  Sorghum bicolor L. Moench 

L12 adverse environment such as what? (i.e., adverse effects such as drought or drought)

L17 Have you refer PLS here before?

L29 Again, please mention full name first.

L33 As a pioneer study of hyperspectral reflectance in plant sciences, it would be good to justify why you chose Sorghum (e.g., importance crop in Korea or Africa?, commonly used crop for PLS analysis or other reasons).

In addition, it is likely to be required more explanation in introduction part of the vegetation index NDVI for readers.

L34 All over the world or specific continent? 

L36 Please simplify this sentence (e.g. It is considered as an important crop in arid and semi-arid areas in West Africa due to-------).

L53 Can you please provide some more references by each parameter?

L61 Again, it would be nice if it is mentioned each parameter with each references, or [17] said PLS method can be used for these 4 factors?

L63 To be specific?

L63 Reactive Oxygen Species? please mention full name first before using abbreviation.

L67 Hydrogen peroxide (H2O2) (if you will use abbreviation next)

L67 oxidative stress?

L69 I did not understand it. 

L71 Too vague. Abiotic stresses such as drought?

L72 Any references?

L74 Can you please rewrite this sentence? 

L75 1)

L76 2)

L83 Full name (Partial least squares regression)

L117 I think it might be easy to provide one table for Figure 1–3 and the correlation graph can be provided in Appendix. 

L122 35 vegetation indices (see Table 1)

L125 Please clarify this sentence.

L127 DAT

L127 significant

L135 It is hard to detect the difference between the two legends. 

L169 I think the discussion should be written by sub-section such as discussion by each result. 

L250 Can you please provide the experimental period and days?

L253 Can you please provide some reference of using medium drought with 60% FC?

L255 What do you mean? How often and how many times has it been irrigated based on FC?

L258 Sorry I did not get it.

L277 How many samples?

L280 How many samples and how many times was it measured? just one time with 4 seedlings in autolog?

L282 Same question with chlorophyll fluorescence and photosynthesis. 

L286 I think Table 1 and 2 can be embedded in Appendix and can provide an experimental design map here. 

I believe the quality of English is fine but can be required some proofreading.  

Author Response

Dear reviewer

Thanks for the valuable comments.

We added explanations related to hydrogen peroxide to the discussion. In addition, relevant references have been added. We also added explanations and information on the vegetation indices.

We changed the title based on your comment.

We adopted abbreviations properly and revised the ambiguous sentences, and wrote them.

We added missed citations properly.

We changed the legend of the Figure you mentioned.

The duration of the experiment was specified.

We described the relationship between field capacity and soil moisture conditions in our experiments in M& M.

Added the number of measurements in materials and methods.

Table 1 has been left, and Table 2 has been converted into an appendix.

Thank you again for your comments that made the thesis more complete.

Round 2

Reviewer 3 Report

The authors tried to improve the quality of the manuscript and mostly revised it based on my comments, but there are still some minor things required. Except for that, I believe there are no additional comments from mine. Please find the attachment. 

Author Response

Dear reviewer

Thank you for your comments

I have revised those parts.

Your comments make the manuscript more scientific.

Thank you again